# Community Integration and Associated Factors 10 Years after Moderate-to-Severe Traumatic Brain Injury

**DOI:** 10.3390/jcm12020405

**Published:** 2023-01-04

**Authors:** Juan Lu, Mari S. Rasmussen, Solrun Sigurdardottir, Marit V. Forslund, Emilie I. Howe, Silje C. R. Fure, Marianne Løvstad, Reagan Overeem, Cecilie Røe, Nada Andelic

**Affiliations:** 1Department of Family Medicine and Population Health, Division of Epidemiology, Virginia Commonwealth University, Richmond, VA 23298, USA; 2Institute of Health and Society, Research Centre for Habilitation and Rehabilitation Models & Services (CHARM), Faculty of Medicine, University of Oslo, 0318 Oslo, Norway; 3Department of Physical Medicine and Rehabilitation, Oslo University Hospital, 0424 Oslo, Norway; 4Institute of Rehabilitation Science and Health Technology, Faculty of Health Sciences, Oslo Metropolitan University, 0130 Oslo, Norway; 5Centre for Rare Disorders, Oslo University Hospital, 0424 Oslo, Norway; 6Department of Psychology, Faculty of Social Sciences, University of Oslo, 0317 Oslo, Norway; 7Research Department, Sunnaas Rehabilitation Hospital, 1453 Bjørnemyr, Norway; 8Faculty of Medicine, Institute of Clinical Medicine, University of Oslo, 0316 Oslo, Norway

**Keywords:** traumatic brain injury, rehabilitation, community integration, CIQ, long-term outcome

## Abstract

This study evaluated the impact of baseline injury characteristics and one-year functional level on the 10-year community integration outcomes for working-age patients with moderate-to-severe traumatic brain injury (TBI). Patients aged 16–55 and diagnosed with moderate-to-severe TBI within 24 h of injury were eligible for the study. Multivariable hierarchical linear regression was utilized to assess the impact of baseline characteristics and one-year functional measures on the mean Community Integration Questionnaire (CIQ) scores 10 years after injury. Of 133 original study participants, 97 survived 10 years, and 75 were available for this study. The mean total CIQ score changed positively from one to 10 years post-injury, from 18.7 (±5.5) to 19.8 (±4.8) (*p* = 0.04). The results suggested that age (β = −0.260, *p* = 0.013), FIM-Cognitive subscale (β = 0.608, *p* = 0.002), and the bodily pain subscale (BP) (β = 0.277, *p* = 0.017) of the SF-36 were significantly associated with the mean CIQ scores. In conclusion, this study demonstrated improved community integration from one to 10 years in a sample of working-age patients with moderate-to-severe TBI. The findings also showed that age, cognitive function, and bodily pain were significant predictors of long-term community integration, suggesting post-acute rehabilitation should focus on factors related to long-term risk and protective factors to improve long-term outcomes.

## 1. Introduction

Patients who survive moderate-to-severe traumatic brain injury (TBI) often suffer long-term disability, including physical, cognitive, and psychological impairments, and subsequent difficulties integrating into their communities [1,2,3,4,5,6]. Therefore, the primary goal of rehabilitation services for this population should not only focus on functional improvement, but also maximize the ability to integrate into the community during the chronic phase [7,8,9].

A recent conceptual analysis suggested that community integration encompasses a variety of components, including independence, a sense of belonging, adjustment, having a place to live, being involved in meaningful occupational activities, and being socially connected to the community [10]. The cognitive, behavioral, and emotional difficulties often experienced by people with TBI may lead to diminished community integration and negatively affect life satisfaction and the quality of life [1]. Given the broad impact of community integration as an outcome following TBI, numerous studies have assessed factors that may facilitate or hamper community integration to inform improved rehabilitation programs. A Norwegian study followed a cohort of patients with moderate-to-severe TBI for their community integration status up to two years post-injury. The findings suggested that certain baseline sociodemographic and injury characteristics, such as younger age, living with a spouse, less severe injury, a longer length of hospital stay, and receiving rehabilitation during acute and subacute phases, were associated with better community integration outcomes at the two-year follow-up point [7]. Another study from Australia used a multivariate correlation design to identify predictors of community integration status and vocational outcome in 209 patient and caregiver pairs. The results showed that age, disability level, and cognitive function, were significant predictors of community integration status at two to five years post-injury [11].

However, the current knowledge on community integration status in patients with TBI was mainly generated from relatively short-term follow-up studies [12,13,14,15,16]. Research on community integration status beyond five-year follow-ups is still limited. Our research group conducted a longitudinal cohort study and followed a cohort of patients with moderate-to-severe TBI in their functional outcomes and community integration status across 10 years, from acute hospital admission up to 10 years post-injury. Community integration status was measured with the Community integration questionnaire (CIQ) at one, two, five, and 10-year follow-up points. We have previously reported the community integration outcomes in the first five years after injury and its baseline predictors. Our findings showed that community integration status improved across one, two, and five-year follow-up points. Further, the results suggested that marital status, higher education level and employment at the time of injury were associated with better community integration outcomes [8].

As an extension of our previous report, this study further describes community integration status up to 10 years post-injury to better understand the long-term functional recovery for patients with moderate-to-severe TBI. It evaluates the impact of baseline demographic, injury characteristics and one-year post-injury functional levels on the 10-year community integration for working-age patients with moderate-to-severe TBI. This approach aimed to identify factors for which adaptations or facilitation could be beneficial one year after TBI to improve long-term community integration. Based on existing literature, we expected that community integration would improve from one to 10 years and that age and gender, injury severity indices, and physical, cognitive, mental, and social functioning one year after injury would be associated with the level of community integration.

## 2. Material and Methods

### Study Setting and Participants

This longitudinal observational study was conducted in a Trauma Referral Center in Norway’s South-Eastern region. All patients aged 16–55 and diagnosed with moderate-to-severe traumatic brain injury (ICD-10 S06.0–S06.9) within 24 h of injury in 2005–2007 were eligible for the study. Patients were excluded if they had previous neurological disorders, spinal cord injuries, previously diagnosed severe psychiatric or substance abuse disorders, and unknown home address or incarceration. Details of the original study design can be found elsewhere [17,18]. The original study recruited 133 patients between 2005 and 2007; patients were followed up for one, two, five, and 10 years. Since the study admission, 32 patients have died, and four withdrew, leaving 97 survivors. Of these, 75 patients (77%) were available for the current study at 10-year follow-up (Figure 1). No statistically significant differences were found in demographics and injury characteristics between individuals assessed at the baseline and those lost to the 10-year follow-up. The Regional Committee for Medical Research Ethics, South-East Norway, and the Norwegian Data Inspectorate approved the study. All participants gave their written informed consent to participate in the study.

## 3. Measurements

Based on the original design, patients’ demographics and injury characteristics were documented at admission [17]. The demographic information included age in years, sex (males or females), marital status (partnered or single), educational level (≤12 years or >12 years), and employment status (yes or no). The injury characteristics consisted of the cause of injury (traffic accident or other), Glasgow Coma Scale (3–8 (severe) or 9–12 (moderate injury)) [19]. Other injury-related characteristics were modified Marshal CT Score (1–2 (less severe) or 3–6 (more severe injury)) [20], Injury Severity Score (ISS) (score range, 1–75 [best to worst]) [21,22], and total acute hospital length of stay in days. Further, for this study, the patient’s functional status and community integration status were collected at one and 10-year follow-ups, respectively. Functional status was assessed with the Functional Independence Measure motor (FIM-M) (score range, 13–91 [worst to best]) and cognitive (FIM-C) (5–35) subscales [23], and Mental Health (MH), Social Function (SF), and Bodily Pain (BP) subscales (score range, 0–100 [worst to best]) of the Medical Outcomes 36-item Short Form Health Survey [24].

The community integration status was evaluated using the Community Integration Questionnaire (CIQ), which is a 15-item scale comprising home (score range, 0–10 (worst to best)) and social integration (0–12) and productive activity (0–7) domains [25]. The total CIQ score ranges from 0–29 points. In the present study, the internal consistency of the CIQ scale at one- and 10 years post-injury was examined with Cronbach’s alpha and found satisfactory (0.79 and 0.76, respectively) [26].

### Statistical Analysis

Descriptive statistics were used to summarize the baseline characteristics and functional status at one-year follow-up and CIQ scores at 10 years. Multivariable hierarchical linear regression was utilized to assess the association between the pre-selected baseline demographic and injury characteristics, one-year functional status, and the outcome of CIQ scores at a 10-year follow-up. Considering the small sample size, a conservative approach was applied for the number of variables in the final model (nine patients per one variable). The model building was stepwise and started with the inclusion of age at baseline, then adding injury parameters at baseline to adjust for the severity of the injury and functional measures at the one-year follow-up; the parameter estimation and model performance were reported via standardized beta values and R^2^, adjusted R^2^, and change in R^2^, respectively. The pre-selected independent variables were based either on theory or findings from previous studies. Before building the multivariable hierarchical linear regression, model assumption assessment and correlation analysis were performed to rule out potential model assumption violation and collinearity between variables. The baseline GCS was removed from the final model due to its high correlation with the modified Marshall CT classification score to improve the model fit. Due to the small number of participants, a sensitivity analysis through 1000 bootstrap repetitions was performed to assess the final model stability. All analyses were performed using the SPSS 28.0 software. Statistical significance was set to a 5% level.

## 4. Results

Table 1 displays participants’ demographic and injury characteristics at baseline and the functional status at 1-year follow-up. Of eligible participants, the mean age at injury was 30.3 (10.8) years, 76% were males, 52% had education ≤12 years, 61% were single, and 83% were employed at the time of injury. Fifty-nine percent of injuries were caused by traffic accidents, and 41% were due to falls, sports injuries, and assaults. Most patients (68%) had severe TBI (GCS 3–8), while 32% had moderate TBI (GCS 9–12). The mean ISS was 30.0 (13.0), and 52% had Modified Marshall CT Score 3–6. The mean total length of acute hospital stay was 29 (24) days.

At one-year follow-up, the means of FIM-M and FIM-C from all participants were 88.6 (±6.4) and 33.1 (±4.2), respectively. The means of the MH, SF, and BP subscales of the SF-36 were 71.4 (±26.9), 69.9 (±20.4), and 75.8 (±29.4), respectively.

The total CIQ score changed positively from one to 10 years post-injury, from 18.7 (±5.5) to 19.8 (±4.8) (Mean difference = 1.1, 95% CI = −2.06–−0.35; Paired T-test = −2.06; *p* = 0.04). The mean subscale CIQ scores at 10 years were: home integration 6.7 (±2.6); social integration 9.4 (±2.1); and productive activity 3.9 (±1.9).

No statistically significant associations were found between gender (*p* = 0.936), education (*p* = 0.249), injury severity (*p* = 0.672; *p* = 0.520), and CIQ mean score at 10 years. At 10 years, 37 (49%) participants were partnered, and 39 (52%) were employed. Of these, 22 (56%) were in the full job. The proportion of persons employed at 10-year follow-up differed significantly by injury severity as 71% of those with moderate TBI were employed compared to 43% of those with severe TBI (*p* = 0.03). Persons with moderate TBI were also more likely to be full-time employed than those with severe TBI, 46% vs. 24% (*p* = 0.04). In total, 53 (71%) of the sample received partly or fully disability pension benefits. A 10-year marital status was not significantly associated with CIQ mean score (*p* = 0.09), but employment status was (*p* = 0.001).

All the one-year functional level subscales were significantly associated with the mean CIQ score at 10 years at *p* < 0.001.

Table 2 displays the multivariable hierarchal linear regression results in describing the association between key characteristics at baseline, functional measures at one-year follow-up, and CIQ scores at 10-year follow-up. Step 1 of the model included participants’ age at injury, step 2 included step 1 plus worst Marshall CT score and ISS at admission, and step 3 further included step 2 plus functional measures at one-year follow-up. The results from the final model suggested that age (standardized β = −0.260, *p* = 0.013), FIM-C (standardized β = 0.608, *p* = 0.002), and BP (0.277, *p* = 0.017) subscales of the SF-36 were significantly associated with the mean CIQ score. The participants’ age at injury was negatively associated with the mean CIQ score. Every unit increase in age was associated with decreasing mean CIQ score while keeping all other variables in the model constant. The FIM-C and BP at 1-year follow-up were positively associated with the mean CIQ score. A unit increase in either cognitive function (FIM-C) or BP (i.e., less pain) was associated with increasing mean CIQ score while keeping all other variables in the model constant. Overall, the final model explained 41% of the variance (R^2^ = 0.408 and adjusted R^2^ = 0.333). More than half of the variance was explained by the functional level at one-year follow-up (change in R^2^ = 0.226). The parameter estimations from the bootstrap analysis supported the estimates in the final model.

## 5. Discussion

This study describes the long-term community integration outcome for patients with moderate-to-severe TBI and evaluates the impact of sociodemographic and functional status post-injury on community integration. Consistent with our findings from the five-year follow-up [8] and the expectations in this study, community integration status had improved slightly in this cohort at the 10-year follow-up assessment but was still below a community-based convenience sample with no history of TBI [25]. In addition to key predictors at baseline, such as age and injury severity measures, functional status one year after injury added substantial weight in predicting community integration outcomes at 10 years. Better functioning, measured as higher scores on the FIM-C and BP subscale of the SF-36, predicted better long-term community integration outcomes. These findings underscore the importance of rehabilitation in addressing patients’ function during the acute and subacute recovery phases of TBI.

The model accounted for 33% of the variance (adjusted R^2^) in CIQ score. One of the reasons why a large proportion of the variance remained unexplained is the limited number of factors that could be used in the model and the multidimensionality of community integration, which is assumed to be influenced by many factors. Some of the remaining explanatory factors may be related to personal factors, such as interpersonal relations [27], environmental conditions, such as the quality of environment [28], or caregiving and current living situation [11].

Cognitive functioning and bodily pain one-year after injury were significant predictors of CIQ, as well as age at the time of injury. Cognitive functioning was the strongest contributor to the explained variance in the CIQ, indicating that higher levels of cognitive functioning in the earlier stages after the injury is a good predictor of better long-term community integration. Our results were consistent with a recent scoping review, which identified cognition as one of the strongest predictors of community integration outcomes following TBI [29]. Further, a previous study on TBI outcomes up to 24 years after injury found that residual cognitive impairments were significantly associated with outcomes such as activity limitations [30]. These findings imply a need for health professionals to address cognitive functioning following TBI, including cognitive rehabilitation programs.

In accordance with previous studies on the trauma population [31], lower bodily pain intensity was significantly associated with better long-term functional outcomes represented by higher long-term CIQ scores. Previous studies have found bodily pain to be common and often persistent among persons with TBI, affecting participation in daily activities and life roles [32,33]. Further, the persistence of pain is an important factor in the disability status of individuals with TBI [34]. Our results indicated that attention should also be paid to pain management in the earlier phases of rehabilitation to maximize the recovery potential and prevent the onset of chronic pain conditions.

Older age is associated with poor outcomes and lower levels of CIQ after TBI [7,8,16,30,35,36,37,38,39,40]. Compared with younger counterparts with TBI, older people are more likely to sustain fall-related TBI [41,42], need longer recovery time, and are less likely to return to work and engage socially after TBI. Moreover, older people are more likely to retire from work after TBI, and older age itself may be associated with decreased social activities due to decreased functions and comorbid conditions other than TBI [40]. This study focused on long-term outcomes for working-age patients with TBI. It demonstrated that age was a significant predictor for the 10-year CIQ score, independent from the injury severity measures and functional status one-year post-injury. Future research should explore the complex interactions between injury and age-related factors and their influence on long-term community integration to better accommodate the rehabilitation needs of this population.

Sex was not significantly related to the CIQ total score at 10 years. One possible explanation is the limited sample variability in sex (76% of patients were males). Marital status, educational level and employment status were significant baseline predictors of five-year CIQ trajectories [8]. However, educational level and marital status at 10 years were not associated with CIQ scores in this study. At the 10-year follow-up, approximately half of the sample was partnered, probably disseminating the effect of the pre-injury marital status as a predictor. This study did not use employment status as a predictor due to the limited variability, as more than two-thirds of participants received work disability benefits at 10 years, and our focus was on functional factors. In line with our results, a scoping review of predictors of community integration following traumatic brain injury by Kersey et al. [29] found less support for demographic characteristics as predictors of community integration, compared to functional and environmental factors. The authors suggested that although previous studies have found demographic characteristics that predict community integration post-TBI, increased research focus on patients’ functioning and environmental factors could explain this finding [29]. Further, we found no significant associations with injury severity variables, suggesting that the influence of injury severity may dissolve over time or that other variables become more important for community integration in the long-term perspective. Kersey et al. [29] found mostly negligible-weak relationships between injury-related variables and community integration outcomes, where the length of post-traumatic amnesia (PTA) was found to have the strongest evidence to support an association. However, we did not use the PTA variable in this study due to the number of missing values. Moreover, the patient’s perception of community integration activities may shift because of the adjustment process to the impairments/disabilities caused by the injury. This may explain the lack of predictive value of injury severity-related variables. A similar explanation may probably be applicable for other non-significant variables from one year, such as mental health and social functioning.

This study is, to our knowledge, one of the first where initial injury severity and early post-injury functioning have been assessed jointly as predictors of community integration at 10 years in a sample of working-age patients with moderate-to-severe TBI. The study findings provide better insight into the evidence of rehabilitation needs after TBI, and the 10-year follow-up period gives valuable knowledge about long-term community integration. However, as an extension of an existing longitudinal study, several study limitations should be acknowledged when interpreting the results. Firstly, the study was conducted in a Trauma Referral Center in Norway’s south-eastern region, where the study setting and the participants’ sociodemographic characteristics may not be generalizable to other populations with TBI. Secondly, the original study was restricted to participants aged between 16 and 55 at admission. Hence, this study’s long-term community integration outcomes were more likely to reflect the results for working-age patients with moderate-to-severe TBI. Thus, the model needs to be validated in future studies against a larger sample of unselected patients with TBI. Thirdly, an inherent small sample size could limit the study’s power and model performance. A further exploration of alternative analytic approaches should be considered in future research, such as principal component analysis, to describe the data structures and their relationship with the study outcomes. Nevertheless, our sensitivity analyses via 1000 bootstraps showed that the estimated direction of associations and *p*-values did not change substantially, supporting robust model estimations.

In conclusion, the current study demonstrated slightly improved community integration from one to 10 years in a sample of working-age patients with moderate-to-severe TBI. The study results also suggested that age, cognitive function, and bodily pain, were significant predictors of long-term community integration for patients with TBI. In post-acute TBI rehabilitation, holistic treatment approaches need to focus on factors related to both long-term risk and protective factors to improve outcomes in the chronic phase.

## Figures and Tables

**Figure 1 jcm-12-00405-f001:**
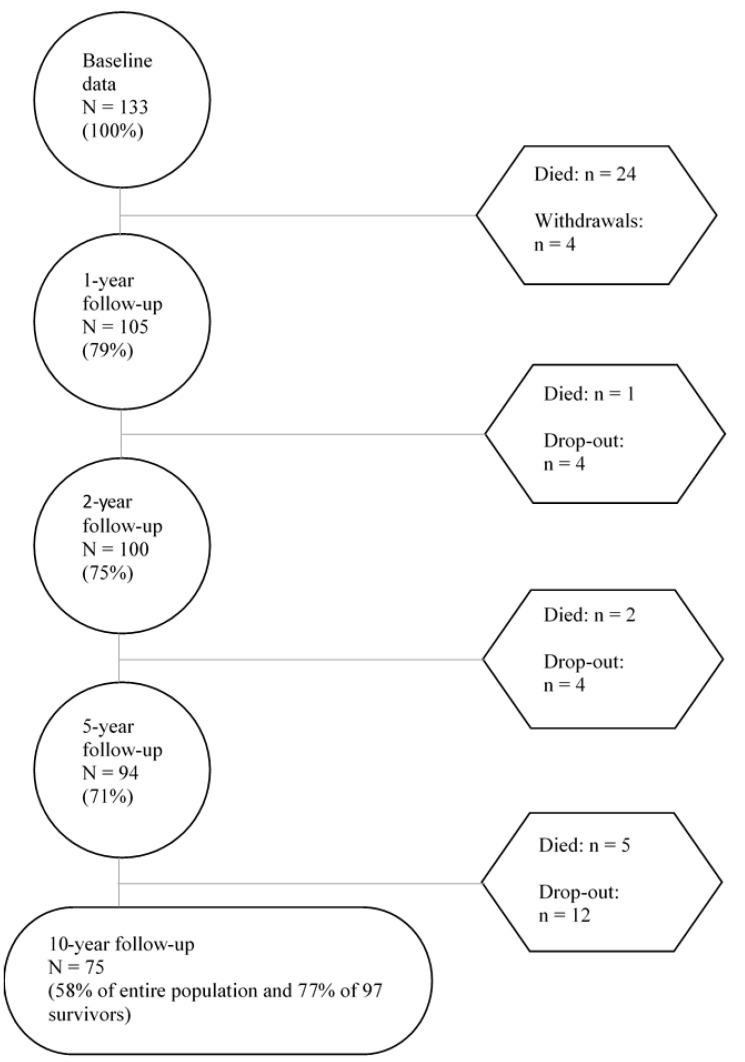
Flowchart depicting baseline and follow-up data through 10-years.

**Table 1 jcm-12-00405-t001:** Participants’ demographic and injury characteristics at baseline and functional status at 1-year follow-up (*n* = 75).

Characteristics	*n* (%)
** *Demographics* **	
Age at injury in years	30.3 (10.8) *
Sex	
Males	57 (76)
Females	18 (24)
Educational Level	
≤12 years	39 (52)
>12 years	36 (48)
Employment status	
Yes	62 (83)
No	13 (17)
Marital status	
Partnered	22 (29)
Living alone	53 (61)
** *Injury characteristics* **	
Glasgow Coma Scale score (GCS)	
9–12 (moderate TBI)	24 (32)
3–8 (severe TBI)	51 (68)
Modified Marshall CT Score	
1–2	36 (48)
3–6	39 (52)
Injury Severity Score (ISS)	30.0 (13.0) *
Total acute length of stay in days	29.0 (24.0) *
** *Functions at 1-year follow-up* **	
FIM-Motor (FIM-M)	88.6 (6.4) *
FIM-Cognitive (FIM-C)	33.1 (4.2) *
SF-36 Social Function (SF)	71.4 (26.9) *
SF-36 Mental Health (MH)	69.9 (20.4) *
SF-36 Bodily Pain (BP)	75.8 (29.4) *

* Mean (SD).

**Table 2 jcm-12-00405-t002:** Results of multivariable hierarchal linear regression: association between baseline characteristics, one-year functional measures and the community integration questionnaire measure at 10-year follow up.

Independent Variables	Model Step 1		Model Step 2		Model Step 3	
	Standardized β-Coefficient	*p*-Value	Standardized β-Coefficient	*p*-Value	Standardized β-Coefficient	*p*-Value
** *Baseline Characteristics* **						
Age	−0.342	0.003	−0.323	0.006	−0.260	0.013
Modified Marshall CT Score			−0.001	0.994	0.110	0.330
Injury Severity Score (ISS)			−0.159	0.192	−0.197	0.083
** *Functions at 1-year follow-up* **						
FIM-Motor (FIM-M)					−0.232	0.191
FIM-Cognitive (FIM-C)					0.608	0.002
SF-36 Social Function (SF)					0.159	0.209
SF-36 Mental Health (MH)					0.201	0.126
SF-36 Bodily Pain (BP)					0.277	0.017
** *Model performance* **						
R^2^	0.117		0.142		0.408	
Adjusted R^2^	0.104		0.104		0.333	
Change in R^2^	0.117		0.025		0.266 *	

* F-test in change = 5.675, *p* < 0.001.

## Data Availability

The data presented in this study are available on request from the corresponding author. The data are not publicly available due to ethical reasons.

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
