# Peer review of "Community Integration and Associated Factors 10 Years after Moderate-to-Severe Traumatic Brain Injury"

_jcm, 2023, doi:10.3390/jcm12020405_

Round 1

Reviewer 1 Report

In the study titled "Community integration and associated factors ten years after moderate-to-severe traumatic brain injury", the authors report the results of a long-term follow-up of patients with traumatic brain injury (TBI), aiming to identify variables associated with better/worse outcomes in terms of community integration.

The highest merit of this study is the really long-term follow-up. Most reports on quality of life or community integration after brain injuries do not provide such a long follow-up. So, this is really good.

However, most of the results are not surprising. In summary, lower age and better functionality are related to better community integration. These results are intuitive, even though I recognize that it is important to demonstrate them objectively.

Therefore, I have 2 main suggestions:

1. The Community Integration Questionnaire (CIQ) score raised from about 19 on 1-year follow-up to 20 on 10-year follow-up. Even though this difference has reaches statistical difference, what is the practical effect on raising 1 single point on CIQ? Do the authors think that this difference implies meaningful difference on patients' life? Or we can say that nothing changes and, therefore, long-term follow-ups will provide the same information as 1 or even 5 years- follow-ups?

2. A factor analysis / principal component analysis (APC) of the subsections of FIM/SF-36 could give more information to the results. The authors found that only FIM-C and SF-36-BP had significant effects on the outcomes. I suggest the authors to perform an APC. Even because the model performance was a low.

Finally, I congratulate the authors on the excellent discussion of the results. They made a god job on discussing their data with important references of the field. The limitations are well exposed and justified.

Author Response

1. The Community Integration Questionnaire (CIQ) score raised from about 19 on 1-year follow-up to 20 on 10-year follow-up. Even though this difference has reaches statistical difference, what is the practical effect on raising 1 single point on CIQ? Do the authors think that this difference implies meaningful difference on patients' life? Or we can say that nothing changes and, therefore, long-term follow-ups will provide the same information as 1 or even 5 years- follow-ups?

Response: We appreciate the reviewer’s comments that the 1-point difference in CIQ score from one year to ten years of follow-up is not of clinical importance. However, we should not overlook that the results are statistically significant and should be reported as such. We, therefore, revised the text in the discussion section. Please see the track and changes for revision on pages 6 and 8.

2. A factor analysis / principal component analysis (APC) of the subsections of FIM/SF-36 could give more information to the results. The authors found that only FIM-C and SF-36-BP had significant effects on the outcomes. I suggest the authors to perform an APC. Even because the model performance was a low.

Response: We agree with the reviewer’s suggestion that APC is one of the formal approaches to exploring the data and identifying underlining trends and gradients in the data structure. However, we also feel that the current study does not have sufficient power to run APC. Various rules of thumb for the required sample size of APC were proposed in the past; one such rule suggests a minimum sample size of 100 and a minimum subject to the variable ratio of 10:1 to 15:1. Given the inherited sample size of 75 participants, our model will likely underfit the data regardless of analytic approaches applied. Therefore, we addressed the reviewer’s comments in the study limitation section. Please see the track and changes on page 8.

Reviewer 2 Report

This study evaluated the impact of baseline injury characteristics and one-year functional level on the ten-year community integration outcomes for working-age patients with moderate-to-severe traumatic brain injury (TBI).

This is a nice and interesting study. Results from this study are not very different from the previous authors' report on the same study population but the authors should be praised for having continued the study up to 10years.

I have very few questions for the authors

1) what were the cause of death? Were they related to the initial trauma? what were the reasons for the drop-out?

2) what was the proportion within each group (severe TBI and mild TBi) for having a partner, employment, having full-time job, and receiving a pension? Indeed, the numbers presented in the results section seem very high for the mild TBI group.

Author Response

1) what were the cause of death? Were they related to the initial trauma? what were the reasons for the drop-out?

Response: We can only report the cause of death in the acute phase while the patient stays at the hospital. So, initial deaths were related to trauma. We could not check the cause of death for those who died after discharge from the hospital. In addition, we are not allowed in Norway to ask participants why they withdraw or drop out of studies. 

2) what was the proportion within each group (severe TBI and mild TBI) for having a partner, employment, a full-time job, and receiving a pension? Indeed, the numbers presented in the results section seem very high for the mild TBI group.

Response: We included moderate-to-severe TBI, but not mild TBI. There were statistically significant differences between the injury severity groups (moderate vs. severe TBI) as assessed by the Chi-square test. See the results from the statistical analysis below. We have now included this in the Result section, page 5. There were no statistically significant differences between the groups about having a partner and receiving a pension at the 10-year follow-up.

10-year follow-up: Crosstab, Chi-square

Having a partner (p=0.225):

Moderate: 10/24= 42%

Severe: 27/51 = 53%

Employment (p=0.025)

Moderate: 17/24= 71%

Severe: 22/51 = 43%

Full-time job (p=0.04)

Moderate: 11/24= 46%

Severe: 12/51 = 24%

Disability pension (p=277)

Moderate: 13/24 = 54%

Severe: 29/51 = 57%

Round 2

Reviewer 1 Report

I am satisfied with the authors' responses.